# Next-Gen brain tumor classification: pioneering with deep learning and fine-tuned conditional generative adversarial networks



Abdullah A. Asiri[1], Muhammad Aamir[2], Tariq Ali[2], Ahmad Shaf[2], Muhammad Irfan[3], Khlood M. Mehdar[4], Samar M. Alqhtani[5], Ali H. Alghamdi[6], Abdullah Fahad A. Alshamrani[7] and Osama M. Alshehri[8]

[1] Radiological Sciences Department, Najran University, Najran, Saudi Arabia
[2] Computer Science, Department of Computer Science, COMSATS University Islamabad, Sahiwal Campus, Sahiwal, Pakistan
[3] Electrical Engineering Department, College of Engineering, Najran University, Najran, Saudi Arabia
[4] Anatomy Department, Medicine College, Najran University, Najran, Saudi Arabia
[5] Department of Information Systems, College of Computer Science and Information Systems, Najran University, Najran, Saudi Arabia
[6] Department of Radiological Sciences, Faculty of Applied Medical Sciences, The University of Tabuk, Tabuk, Saudi Arabia
[7] Department of Diagnostic Radiology Technology, College of Applied Medical Sciences, Taibah University, Madinah, Saudi Arabia
[8] Department of Clinical Laboratory Sciences Faculty of Applied Medical Sciences, Najran University, Najran, Saudi Arabia

Corresponding author
Ahmad Shaf,
ahmadshaf@cuisahiwal.edu.pk

## ABSTRACT

Brain tumor has become one of the fatal causes of death worldwide in recent years, affecting many individuals annually and resulting in loss of lives. Brain tumors are characterized by the abnormal or irregular growth of brain tissues that can spread to nearby tissues and eventually throughout the brain. Although several traditional machine learning and deep learning techniques have been developed for detecting and classifying brain tumors, they do not always provide an accurate and timely diagnosis. This study proposes a conditional generative adversarial network (CGAN) that leverages the fine-tuning of a convolutional neural network (CNN) to achieve more precise detection of brain tumors. The CGAN comprises two parts, a generator and a discriminator, whose outputs are used as inputs for fine-tuning the CNN model. The publicly available dataset of brain tumor MRI images on Kaggle was used to conduct experiments for Datasets 1 and 2. Statistical values such as precision, specificity, sensitivity, F1-score, and accuracy were used to evaluate the results. Compared to existing techniques, our proposed CGAN model achieved an accuracy value of 0.93 for Dataset 1 and 0.97 for Dataset 2.

## INTRODUCTION

In current years many machine learning techniques have revolutionized the field of medical image examination by enabling more accurate and efficient analysis of complex

medical data. These techniques involve using algorithms and computational methods to analyze medical images, such as MRI (*Ghassemi, Shoeibi & Rouhani, 2020*) and CT scans, to assist medical professionals in making diagnoses and treatment decisions. In order to detect and characterize brain tumors, MRI contrast plays a critical role in brain tumor scanning by improving the sensitivity and specificity of MRI scans. It is easier to distinguish between different types of brain tumors and see their location and extent when contrast agents increase the contrast between normal and diseased tissues (*Lundervold & Lundervold, 2019*; *Asiri et al., 2022b*). Some standard image processing methods used in medical imaging include segmentation, registration, and classification, which help extract meaningful information from medical images and improve their interpretability. These techniques have become increasingly important in radiology, where they aid in diagnosing diseases such as cancer and neurological disorders (*Krizhevsky, Sutskever & Hinton, 2017*). Overall, computer vision and image processing techniques have greatly improved the accuracy and efficiency of medical image analysis, resulting in more precise diagnoses and better patient outcomes. Brain tumors can arise from various factors, but they typically result from abnormal cell growth within the brain (*Suzuki, 2017*).

The human brain comprises various cell types, including neurons, glial, and support cells. Brain tumors can originate from any of these cell types, but gliomas are the most common type of brain tumor. These tumors consist of all the glial tissues in the central nervous system (*Louis et al., 2016*). Genetic mutations may be responsible for some brain tumors, but most cases have an unknown etiology. Brain tumors can be categorized as benign (non-cancerous) or malignant (cancerous). Malignant brain tumors are highly aggressive and can invade other regions of the brain or body, making them notoriously difficult to manage (*Abd-Ellah et al., 2019*). The symptoms of brain tumors can vary depending on their location and size but may include headaches, seizures, nausea, and changes in cognitive or motor function. The usually used techniques for treating brain tumors may consist of surgical treatment, radiotherapy, chemotherapy, or a mixture of these methods (*Shibuya, 2015*).

The classification and detection of brain tumors using MRI is a crucial task in medical imaging. One common type of brain tumor that can be identified through MRI is gliomas (*Mortensen et al., 2022*), which originate in the brain's glial cells and can be further subcategorized based on the type of glial cell from which they arise, such as astrocytomas, oligodendroglia, and ependymomas. Meningioma tumors, on the other hand, originate in the meninges surrounding the brain and spinal cord (*Garzon-Muvdi et al., 2020*; *Asiri et al., 2022b*). Finally, pituitary tumors originate in the pituitary gland, which is located within the brain structure (*Ilie et al., 2022*), classified brain tumor on MRI, radiologists look for specific features characteristic of each type of tumor. These features can include the tumor's size, location, shape, and enhancement pattern on contrast-enhanced MRI. The radiologist may also use other imaging modalities, such as CT and PET scans, to help make a more accurate diagnosis. Ultimately, a biopsy may be necessary to confirm the diagnosis of a brain tumor (*Nazir, Shakil & Khurshid, 2021*).

Various deep-learning techniques, including CNNs, have been extensively utilized for brain tumor detection and classification. CNNs are deep learning algorithms that

automatically learn and extract image features. This property makes them highly suitable for tasks such as medical image analysis, where identifying small, subtle patterns and structures is crucial (*Deepak & Ameer, 2019*). However, the detection and classification of brain tumors can be defined as the accuracy of identifying the tumors in brain MRI images to support the early-stage diagnosis and treatment of brain tumors. It involves developing algorithms that can automatically detect the presence of tumors, accurately segment them from the surrounding brain tissue, and classify them into different types and grades based on their characteristics. The goal is to give accurate and timely evidence to the physicians so that they can make precise decisions about patient care and treatment options. This problem statement is critical in medical imaging and can impact patient outcomes and the worth of life (*Gull & Akbar, 2021*; *Almalki et al., 2022*).

*Brunetti et al. (2019)* provides an impression of the medical imaging techniques usually used for the diagnosis of neoplasias, including ultrasounds, computed tomography (CT) (*Scola et al., 2022*), MRI (*Anaraki, Ayati & Kazemi, 2019*), and positron emission tomography (PET) (*Lohmann et al., 2019*). There are the numerous types of neural networks that have been applied to the classification task, including convolutional neural networks (CNNs) (*Gómez-Guzmán et al., 2023*), recurrent neural networks (RNNs) (*Maqsood, Damaševičius & Maskeliūnas, 2022*), and deep belief networks (DBNs) (*Kharrat & Néji, 2019*).

To achieve the goal of MRI brain tumor detection and classification, a conditional generative adversarial network (CGAN) is employed. The CGAN model comprises two neural networks: the generator and the discriminator. The generator network is trained to generate images similar to real MRI images, while the discriminator network distinguishes between authentic and generated images. The value of the CGAN model is then used as input for fine-tuning a CNN model, which generates authentic images with corresponding labels. The use of CGANs on magnetic resonance images (MRIs) creates several contributions, including:

- CGAN represents a significant improvement over traditional machine learning algorithms and can help to improve the accuracy of diagnosis, brain tumor detection and classification.
- CGAN reduces the potential for human bias in tumor diagnosis by using an automated system; the potential for human error or subjective interpretation can be minimized.
- CGAN generates realistic tumor images that can help improve the model's interpretability. It can provide additional insight into the underlying biology of the tumors and aid in developing more targeted and effective treatments. Overall, using CGANs on MRIs can improve the accuracy and speed of diagnosis, reduce human bias, and provide new visions into the underlying biology of brain tumors.

The remaining article is prepared as the introduction summarizes the research problem and its significance, setting the stage for the study. The literature review presents a comprehensive summary of previous research and theories that inform the current study. The methods section describes the study design, including the participants, data collection

procedures, and statistical analyses. The results section explains the study's outcomes, including any tables or figures that help illustrate the data. The discussion section interprets the results, discussing their implications for the research question and identifying study limitations. Finally, the conclusion summarizes the main findings and their significance, highlighting any implications for future research.

## STATE-OF-ART RELATED WORK

In the last few years, various deep learning and traditional learning techniques have been introduced for the image-based analysis of brain tumors. While many of these techniques have been used as benchmarks for accurate detection and classification of brain tumors, some recent approaches have also shown promise. The following section discusses some of the latest approaches in the related work.

The conditional generative adversarial networks (CGANs) (*Rezaei et al., 2018*) could also be used for brain tumor detection and classification by training the network to generate realistic tumor images based on input MRI data. The CGAN contains a generator model and a discriminator model, which are trained in an oppositional manner. The first one is trained to create realistic tumor images. On the other hand, a discriminator network works to discriminate between real tumor images and fake tumor images generated by the generator network. However, it does not extensively address the computational complexity and resource requirements associated with training and deploying CGANs for brain tumor detection and classification.

In CGANs, brain tumor detection and classification, the input MRI data is fed into the generator network, which generates a tumor image based on the input data. The generated tumor image is then compared to the ground truth image to determine the accuracy of the generated image. The discriminator network can also classify the generated image as tumor or non-tumor. Generating and classifying tumor images can be repeated iteratively to enhance classification accuracy (*Rezaei, Yang & Meinel, 2019*). Whereas, it does not extensively explore the potential challenges and complexities associated with iteratively generating and classifying tumor images, which could impact the efficiency and practicality of the method in clinical applications.

In *O'Reilly et al. (2019)*, the author proposes a new method for segmenting brain images into regions that may contain cancerous cells. The authors introduce a cellular automata (CA) model to simulate the spread of cancerous cells and use this model to guide the segmentation process. The proposed method extracts the region of interest from an MRI brain image and then applies a Gaussian filter to smooth the image and reduce noise. Next, the initial cancer seed is identified based on the intensity thresholding. The CA model is used to simulate the propagation of the cancerous cells from the initial seed, with the propagation rule defined based on the intensity values of neighbouring pixels. The simulation is iterated until a steady state is reached, and the resulting simulated image is used as a guide to segment the original image. The proposed technique was tested on a dataset of MRI images, and the results were compared with several other segmentation methods. The author reports that their method achieved higher accuracy and better dice similarity coefficient (DSC) than the other methods. This simulation-based approach have

limitations in accurately representing all possible variations of cancerous regions, potentially leading to segmentation errors in certain cases.

The CGANs are trained on a brain MRI dataset with labelled tumor regions to learn the relevant features for tumor detection. The trained CGANs can then classify new brain MRIs as tumor or non-tumor (*Ding et al., 2021*). Authors assumes that the training data used for CGANs is representative of all possible variations and characteristics of brain tumors, which may not fully account for the heterogeneity of real-world tumor images.

Another three-dimensional conditional generative adversarial network (CGAN) (*Ghaffari, Pawar & Oliver, 2021*) replicated for reducing motion objects in brain MRI has been proposed. It consists of two stages: a simulation stage, where motion is added to the brain MRI data to create a motion-corrupted dataset and a reduction stage, where the 3D CGAN is trained on the motion-corrupted dataset to create motion-free images. The proposed method provides a promising approach for reducing motion artifacts in brain MRI, which could enhance the accuracy of diagnosis and action planning for brain disorders. While this is beneficial for improving image quality and diagnosis accuracy, it didn't address other potential sources of image artifacts or distortions, such as hardware-related issues or patient-specific factors.

Another study (*Conte et al., 2021*) proposes a novel approach for the classification of brain tumors based on MRI images using a deep neural network with trained generative adversarial networks (GANs). It relies on the availability of a large labeled dataset for training both the generative adversarial networks (GANs) and the deep neural network (DNN). Gathering such labeled data, especially in the medical field, can be challenging and time-consuming.

*Ahmad et al. (2022)* The proposed method involves two stages: a pre-training stage and a fine-tuning stage. In the pre-training stage, a GAN generates realistic tumor images to pre-train the DNN. Then, in the fine-tuning stage, the pre-trained DNN is further trained using a labelled dataset of brain MRI images for classification. The method's performance is evaluated on a dataset of MRI images with four different types of tumors, and the results show its potential to aid medical professionals in tumor treatment. Using a DNN with GAN pre-training, this approach presents promising outcomes for diagnosing tumors based on MRIs.it involves a two-stage training process, which can be computationally expensive and time-consuming. Additionally, the success of the method relies on the quality of the GAN-generated tumor images, which is challenging to control and optimize.

The article (*Asiri et al., 2022a*) proposes a block-wise neural network (BWN) for brain tumor diagnosis in MRI. The proposed method divides the MRI into several blocks and uses a DNN to classify each block as tumor or non-tumor. The BWN is trained using a brain MRI dataset with labelled tumor regions, and the performance is evaluated using several statistical metrics like accuracy. The results reveal that the anticipated BWN method attains high accuracy and AUC compared to other existing MRI methods for brain tumor identification. The authors also examine the performance of the BWN in the form of different hyperparameters and provide insights into the learned features and their relevance to the classification task. The proposed BWN approach provides a promising method for brain tumor identification in MRI, which could assist medical professionals in

investigating and curing brain tumors. While the BWN approach shows promise for brain tumor identification in MRI, addressing the spatial dependencies between image blocks, providing guidance on hyperparameter tuning, and reporting a broader set of evaluation metrics would contribute to a more thorough understanding of its strengths and limitations.

Conversely, a multi-level deep generative adversarial network (MDGAN) (*Asiri et al., 2023*) for brain tumor diagnosis on MRI consists of a series of pre-trained DNNs with supervised and unsupervised learning. The DNNs are trained to produce realistic tumor images to pre-train the classifier, which is then fine-tuned using a labelled brain MRI dataset. The authors also analyze the results of the method in the form of different hyperparameters and provide insights into the learned features and their relevance to the classification task. The proposed MDGAN approach provides a promising method for brain tumor classification based on MRIs. Due to the involvement of multiple pre-trained DNNs with supervised and unsupervised learning stages and the complexity can be advantageous for capturing intricate features in brain tumor images, it may also pose challenges in terms of computational resources and training time.

## PROPOSED METHODOLOGY

The section on methodology presents the proposed approach that involves using a CGAN comprising two models, namely, the generator and the discriminator. A detailed explanation of the methodology adopted is given below.

### Generator model

The GAN is a deep learning model comprising two neural networks: the generator and the discriminator. The generator takes a random input vector and produces a real training data sample. In contrast, the discriminator obtains genuine and fake data samples from the generator and attempts to differentiate between them. In a conditional GAN, an additional input variable, such as class labels or attributes, is applied as a conditional vector to generate samples that belong to a specific class or possess specific attributes.

For incorporating the conditioning input, it is concatenated with the random noise vector at the input layer of the generator model. The concatenated input is then passed through multiple layers of neural networks to transform it into an output that mimics the target data distribution. During training, both models are trained adversarial, where the discriminator model tries to distinguish between the realistic and fake data samples, and the generator tries to generate samples that cheat the discriminator into relying on they are genuine. The two models are iteratively trained until the generator creates vague samples from the real data samples according to the discriminator, as shown in Fig. 1.

The generator model is a crucial component of GANs, and its primary objective is to generate synthetic data that closely resembles the training data. The generator model's behavior depends on the type of model used and the data it is trained on. In the case of GANs, the generator model generates synthetic data, such as images or videos, similar to the real data used for training. The generator model is typically trained alongside a discriminator model, differentiating between real and synthetic data. Therefore, the

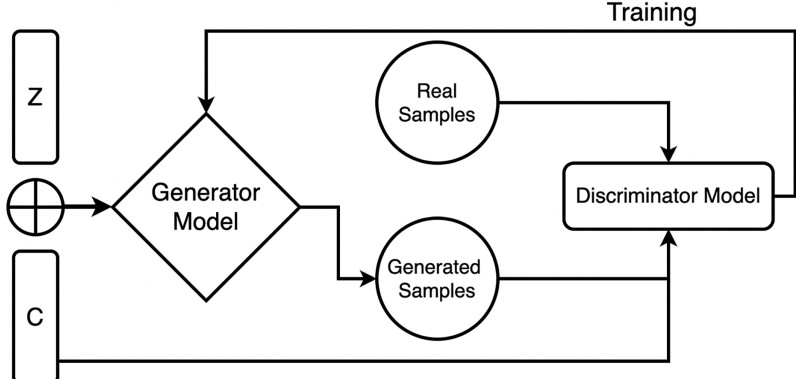

**Figure 1 Generator and discriminator model working in proposed CGAN.**

generator model aims to create synthetic data that appears genuine and can deceive the discriminator model.

The generator model learns patterns and structures in the training data that it can utilize to generate new data. It operates similarly to a typical CNN. In this regard, the design and working of the CNN model are unique. The generator model employs seventeen layers, and the first layer's input can be identified and its image size determined by adjusting the channel size, height, and weight.

In summary, a Generator Model's overall behavior is to create new data that closely resembles the training data it was trained on. It accomplishes this by learning the patterns and structures in the training data that it can use to generate new data.

To improve the speed of the network, the framework incorporates an additional layer of batch normalization. Furthermore, the ReLU activation function is utilized. To determine each batch's highest value, and the max-pooling layer splits the input into multiple pooling sections. The network was trained using a learning rate (Lr) of 0.01 and an epoch value of 30. Various constraints and kernel (K) dimensions ranging from 2 to 8 are also utilized. After the pooling layer, the fully connected layer is introduced, which combines all the outputs from the preceding layers. It connects the learned characteristics of the layers to identify more extensive configurations. The softmax function serves as a normalizing activation function, and the categorization layer uses probabilities to classify the output into predefined categories.

In other words, an extra layer of batch normalization is introduced to improve the network's speed, and the ReLU activation function is implemented. The max-pooling layer is used to compute the maximum value for each region, and a range of constraints and kernel dimensions are applied. The fully connected layer connects the learned features of the layers to recognize more complex patterns. Finally, the softmax activation function is used to normalize the output, and the classification layer is responsible for categorizing the output into designated categories based on probabilities.

$$FR = 0, for \quad x < 0 \quad and \quad 1, for \quad x \geq 0 \tag{1}$$

$$F^c = I^f_c(fi^{c-1}) = (fi^{c-1})^t W^c_i + T^c \qquad (2)$$

$$Con = \frac{(R_d + 2P_a - F_d)}{S_{tr}} + 1 \qquad (3)$$

F (R) indicates the ReLU function, and x is the target value, as shown in Eq. (1).

Equation (2), display the fully connected layer output, f filter number, I as input, W indicates the weight, and T is the constant value.

The Con shows the convolutional layer, is the input dimension, indicates the padding, shows the filter's dimensions, and is the stride in Eq. (3).

## Discriminator model

The discriminator model makes up the second half of the CGAN model. In CGAN, the discriminator model is in charge of separating genuine data from fake data and assessing if the fake data produced by the generator model satisfies the requirements set by the input to the network. Real and artificial data and their related condition vectors are all inputs to the discriminator model in a CGAN. Labels, attributes, and any other pertinent information that can be utilized to produce particular sorts of data are included in the condition vectors as extra information about the data.

The discriminator model then processes the input data and condition vectors to yield a single scalar value representing the likelihood that the input data is real or fake, given the associated condition vector. It is trained to output a high probability of accurate data with matching conditions and a low probability of false data with mismatched or wrong conditions as shown in Table 1.

The discriminator model is trained to achieve the highest level of accuracy in identifying authentic and fraudulent data under matching conditions. The discriminator model can be tricked into thinking that bogus data is real with matching criteria by training the generator model to produce false data. It establishes a feedback loop in which the generator model improves at producing realistic data that match the criteria. In contrast, the discriminator model improves at differentiating between actual and fake data.

In a CGAN, the discriminator model is crucial since it advises the generator model on producing more realistic data while guaranteeing that the data matches the requirements. The discriminator's function is employed to validate the artificially manufactured output of the generator. The discriminator model predicts a true or false binary class label with a domain sample as input. For developing a discriminator, a standard categorization model is used. The discriminator model is dropped since the generator, after training, reveals the exciting space. Since the generator has mastered separating characteristics from instances in the problematic domain, it can occasionally be reused.

It works like a conventional deconvolutional neural network. There are ten layers used in this model with different parameters. The input layer, the top layer, applied some limitations to the channel's width, height, and weight. The kernel (K) values vary from 2 to 8. Also, with an epoch value of 30, the training procedure's learning rate (Lr) is 0.001.

**Table 1 Steps for Pseudo code of CGAN model.**

Input: Variable vectors as R(i) real image, N(i) noise

Output: F (N(I)) indicate fake image

1. For iteration_num do

2. For step_num(S) do

3. Fake images generation F (N (1)),..., F (N (m))

4. Result as a real images {R(1),..., R(m)}

5. Discriminator (D) model training:

$\Delta\theta dm[\log D(Ri) + \log(1 - D(F(Ni)))]$

6. end for

7. Fake images generation {F(N(1)),..., F(N(m))}

8. Generator (G) model training: $\Delta\theta gm \log(1 - D(G(Ni)))$

9. end for

Certainly, let's break down the steps outlined in the pseudocode for the CGAN model:

Input: Variable vectors as R(i) real image: These are real images from the dataset, denoted as R(i), where 'i' represents the index of the image.

N(i) noise: These are noise vectors, denoted as N(i), which serve as random input to the generator network for generating fake images.

Output: F(N(i)) indicates fake image: These are the generated fake images produced by the generator network when given the noise vectors N(i).

1. For iteration_num do: This indicates that the following steps will be repeated a certain number of times, specified by the 'iteration_num'.

2. For step_num(S) do: Within each iteration, there is another loop, which repeats for a certain number of steps, specified by 'step_num' denoted as 'S'.

3. Fake images generation F(N(1)),..., F(N(m)): In this step, the generator network (G) takes a set of noise vectors (N(1), N(2),..., N(m)) as input and produces a corresponding set of fake images (F(N(1)), F(N(2)),..., F(N(m))). Each 'N(i)' is used to generate a corresponding 'F(N(i))'. This step generates fake images from noise.

4. Result as real images R(1),..., R(m) : In parallel to generating fake images, this step represents real images from the dataset (R(1), R(2),..., R(m)), where each 'R(i)' corresponds to a real image. These real images are used for comparison during the training of the discriminator network.

5. Discriminator (D) model training: In this step, the discriminator network (D) is trained. The training process involves optimizing the discriminator's parameters denoted as $\theta dm$ to minimize the following loss function:

$\Delta\theta dm[\log D(Ri) + \log(1 - D(F(Ni)))]$

Here, 'log D(R_i)' represents the discriminator's prediction for real images, and 'log(1-D(F(N_i)))' represents the prediction for fake images. The discriminator aims to correctly distinguish real images from fake ones.

6. end for: This marks the end of the loop for training the discriminator.

7. Fake images generation F(N(1)),…, F(N(m)): After training the discriminator, the generator network (G) is used again to generate another set of fake images from the same noise vectors.

8. Generator (G) model training: In this step, the generator network (G) is trained. The training process involves optimizing the generator's parameters denoted as $\theta\_gm$ to minimize the following loss function: $\Delta\theta gm \log(1 - D(G(Ni)))$

The generator aims to produce fake images that are convincing enough to fool the discriminator. The loss function encourages the generator to generate images that the discriminator is likely to classify as real.

9. end for: This marks the end of the loop for training the generator.

These steps represent the core training process of a CGAN. The generator and discriminator networks are trained iteratively, with the discriminator trying to distinguish real from fake images, and the generator trying to produce fake images that are realistic enough to fool the discriminator. This adversarial training process continues for the specified number of iterations, improving the quality of the generated images over time.

## Why choose sequential model

The choice of a sequential model in the CGAN study is motivated by the nature of the input data and the requirements of the conditional generative adversarial network architecture. In the context of image generation and processing, particularly in medical image analysis like brain tumor detection, sequential models offer several advantages:

**Handling sequential data:** Medical images, including MRI scans, are inherently sequential in nature, as they consist of a series of two-dimensional slices or frames. A sequential model, such as a convolutional recurrent neural network (CRNN) or a similar architecture, is well-suited to handle such sequential data. It can capture spatial dependencies within individual images (through convolutional layers) while also considering the temporal dependencies between images in a sequence (using recurrent layers). This is crucial for understanding the context and structure of the data.

**Contextual information:** Brain tumor detection in MRI scans often requires understanding the context of each image within a sequence. For example, the appearance of a tumor may change across different slices, and recognizing these changes is essential for accurate diagnosis. A Sequential Model helps in capturing this contextual information, enabling the generator to generate realistic images that maintain consistency across the sequence.

**Conditional generation:** In the context of a CGAN, the generator produces images conditioned on some input data, typically noise or other images. When generating images, it's essential to consider both the content of the input data and the sequential context. For example, in medical image synthesis, the Sequential Model can be used to ensure that generated images align with the anatomical structures seen in previous slices.

**Model flexibility:** Sequential models provide flexibility in designing the architecture to suit the specific needs of the task. They allow for the incorporation of both convolutional layers for spatial feature extraction and recurrent layers for temporal modeling. This

flexibility is advantageous for tasks like brain tumor detection, where both spatial and temporal information is crucial.

In summary, the choice of a sequential model in the CGAN study is driven by the need to effectively capture spatial and temporal dependencies in medical image data, ensure conditional generation that respects the sequential context, and ultimately improve the quality and realism of generated images. This choice aligns with the specific requirements of the brain tumor detection task in MRI scans.

## Model working

The pre-trained CNN model discriminates in CGAN to distinguish between the true MRIs generated by the generative model and the real ones, allowing the discriminator to extract and learn MRI features. Subsequently, the pre-trained CNN is used to final classify brain tumors. The final fully connected net in the classification module is replaced with a softmax layer to improve classification accuracy. To clarify our methodology, we fine-tuned a pre-trained conditional generative adversarial network (CGAN) to generate realistic brain tumor images. Then, we extracted features from the discriminator network using the pre-trained weights and fed them into a new model for classification. To do this, we created a new sequential model and added the layers of the discriminator up to the last layer, excluding the final classification layer. We froze the layers of the discriminator to ensure that the pre-trained weights were not modified during the training of the new model. Then, we added a new dense layer with two output nodes and a softmax activation function for binary classification. The summary of the new model is as follows:

1. new_model = Sequential().
2. for i in range(len(disc.layers)−1):
3. new_model.add(disc.layers[i]).
4. Freeze the layers.
5. for layer in new_model.layers:
6. layer.trainable = False.
7. new_model.add(Dense(classes,activation='softmax')).

In contrast, a few others are generated using a generator model that takes a randomized vector as input from a specific latent space. This model produces some sample images as output. The discriminator takes the real image as input, deconvolutes it and then provides a binary classification for real and fake images during training and testing.

## RESULTS AND DISCUSSION

To evaluate the results, Python (version 3.6.6) was used to implement all models in this research, along with various supporting libraries, including NumPy (np), Keras models utils, matplot library, and Scikit-learn. The applied model was computed on a system equipped with a Core i7 10th generation processor, Nvidia graphics card of 6 GB, and RAM of 16 GB.

## Dataset details and preprocessing

The brain MRI images tumor dataset used in this study is publicly available on Kaggle, Dataset 1: (https://www.kaggle.com/datasets/navoneel/brain-mri-images-for-brain-tumor-detection). Dataset 2: (https://www.kaggle.com/datasets/navoneel/brain-mri-images-for-brain-tumor-detection) Both datasets consist of two classes: "no," which indicates the absence of a brain tumor, and "yes," which indicates the presence of a brain tumor. Dataset 1 consists of a total of 239 images, with 84 for "yes" and 155 for "no" classes; on the other hand, dataset 2 consists of a total of 3,000 images, with 1,500 for "yes" and 1,500 for "no". The dataset contains images of various dimensions, and sample images are shown in Fig. 2. To standardize both datasets, duplications are removed, and all images are resized to 128 × 128 resolution. The datasets are then thoroughly cleaned by removing duplicate values, missing labels, and extensions. Additionally, a histogram equalizer removes any noise from the images. The datasets are divided into 80% training and 20% testing sets for experimentation. Further details about the dataset are provided in Table 2.

## Evaluation criterion

The proposed brain tumor classification and detection CGAN network employs several arithmetic calculations outlined in Eqs. (4)–(7). The term accuracy is evaluated by counting the correctly classified positive images (Tp) and negative images (Tn), while positively incorrect classified images are labeled as false positives (Fp). The negatively classified images total images are represented by false negatives (Fn). The statistical (Eqs. (4)–(7)) are necessary to determine these values:

$$Precision = \frac{T_p}{T_p + F_p} \tag{4}$$

$$Recall = \frac{T_p}{T_p + F_n} \tag{5}$$

$$F1 - score = \frac{2T_p}{2T_p + F_p + F_n} \tag{6}$$

$$Accuracy = \frac{T_p + T_n}{T_p + T_n + F_p + F_n} \tag{7}$$

## Model results

The model generates fake images during training, as shown in Fig. 3. The process works in two stages: CGAN generates the fake images, and then the CNN model takes them as input to classify them as fake or original images based on the exact labels. The confusion matrix for the fake image generation on the test dataset is shown in Fig. 4, where the dataset consists of two classes: Yes and No. The Yes class indicates the occurrence of a brain tumor, while the No class indicates no tumor occurrence detected by the model. Additionally, Fig. 5 illustrates the training and validation accuracy and loss of the applied

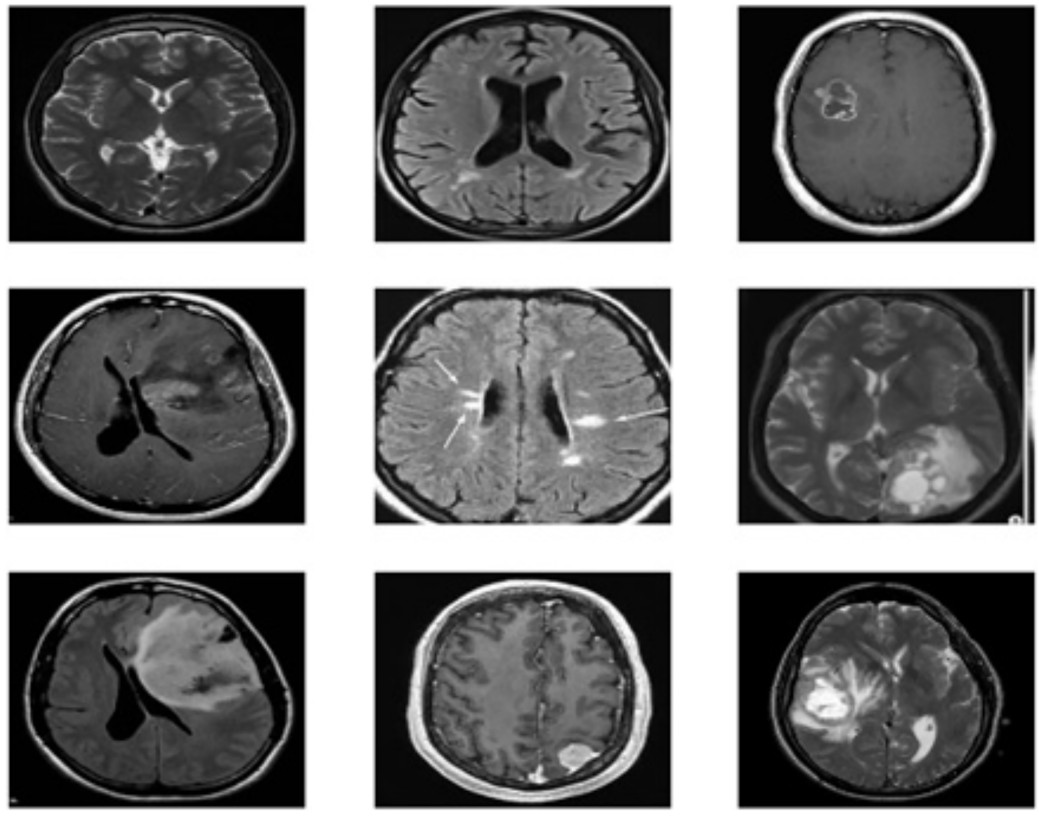

**Figure 2 Sample images of brain tumor dataset.**

**Table 2 Distribution of data in Dataset 1 and Dataset 2.**

| Classes | Dataset 1 | | | Dataset 2 | | |
| --- | --- | --- | --- | --- | --- | --- |
| | Images | Train | Test | Images | Train | Test |
| No | 84 | 66 | 18 | 1,500 | 1,200 | 300 |
| Yes | 155 | 135 | 20 | 1,500 | 1,200 | 300 |
| Total | 239 | 201 | 38 | 3,000 | 2,400 | 600 |

CGAN model. The blue dotted line represents the training accuracy and loss, while the red one shows the validation accuracy and loss.

Figure 6 compares the generator and discriminator performance on Datasets 1 and 2, measured in terms of loss concerning batch number. Dataset 1 consists of 201 images, and dataset 2 takes 2,400 images of brain tumors and was used to train the GAN model. The training was conducted using a batch size of 8 and 500 epochs with 100 noise dimensions. In total, 12,000 and 140,000 batches were produced throughout 500 epochs in dataset 1 and 2.

The graph shows the maximum loss captured from batches 0 to 6, indicating that both the generator and discriminator improve over time as the number of training epochs increases. The loss of the generator decreases steadily over time, while the loss of the
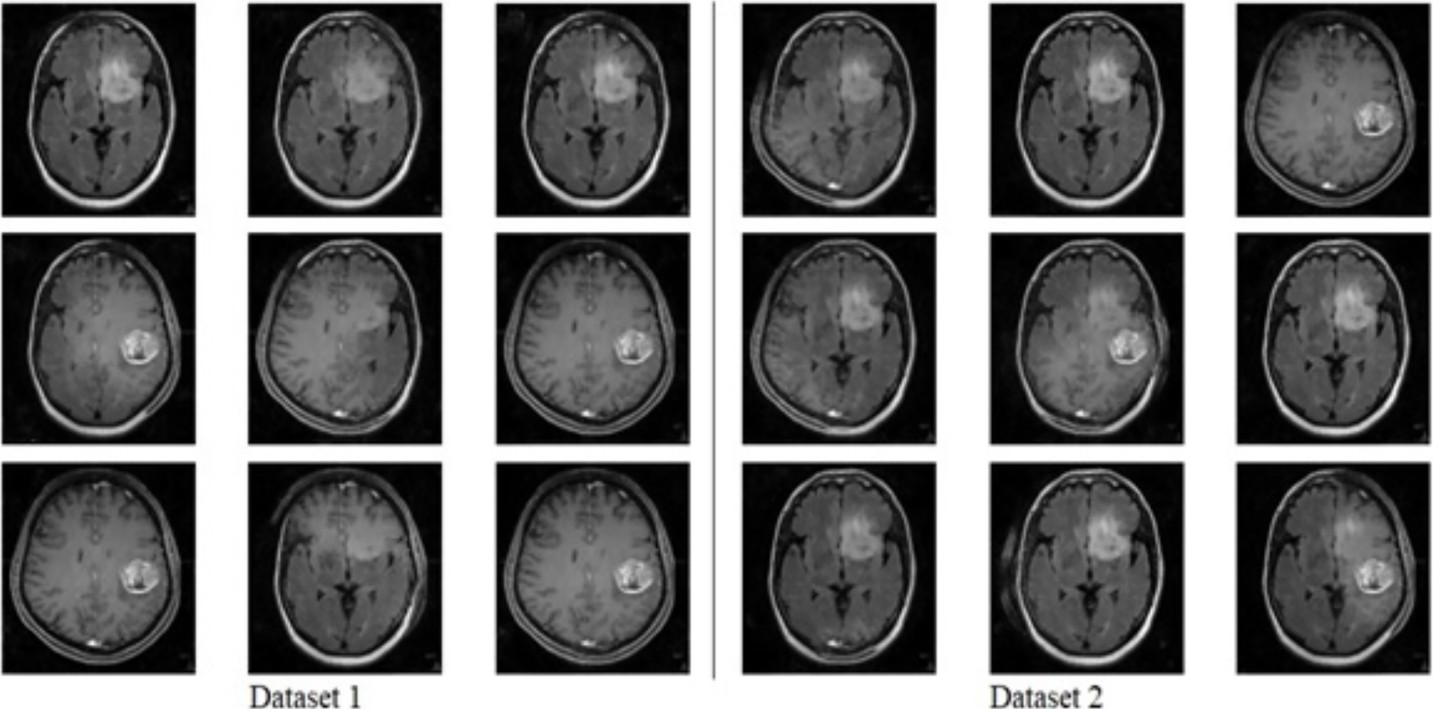

**Figure 3 Fake image generated by CGAN on Dataset 1 and Dataset 2.**

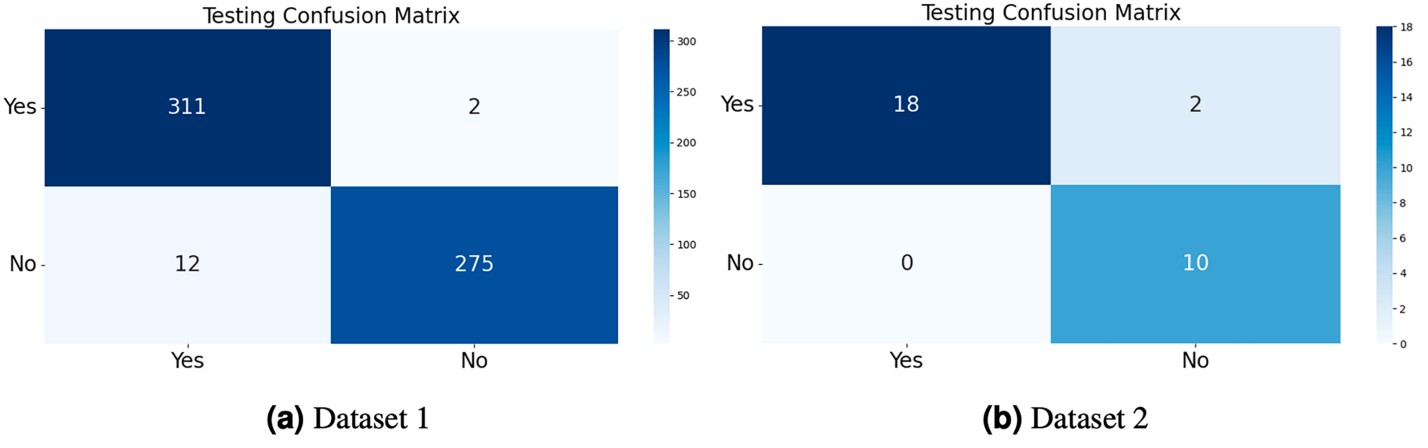

**(a)** Dataset 1                     **(b)** Dataset 2

**Figure 4 The confusion matrix of the test dataset was computed for both Dataset 1 and Dataset 2, allowing for an evaluation of the accuracy, precision, recall, and F1-score of the proposed CGAN model for the classification of brain tumors.**

discriminator fluctuates but remains relatively stable overall. These results suggest that the GAN model is effectively learning to generate more realistic images of brain tumors over time. Overall, the graph provides insight into the training process and the performance of the GAN model on both Dataset 1 and Dataset 2.

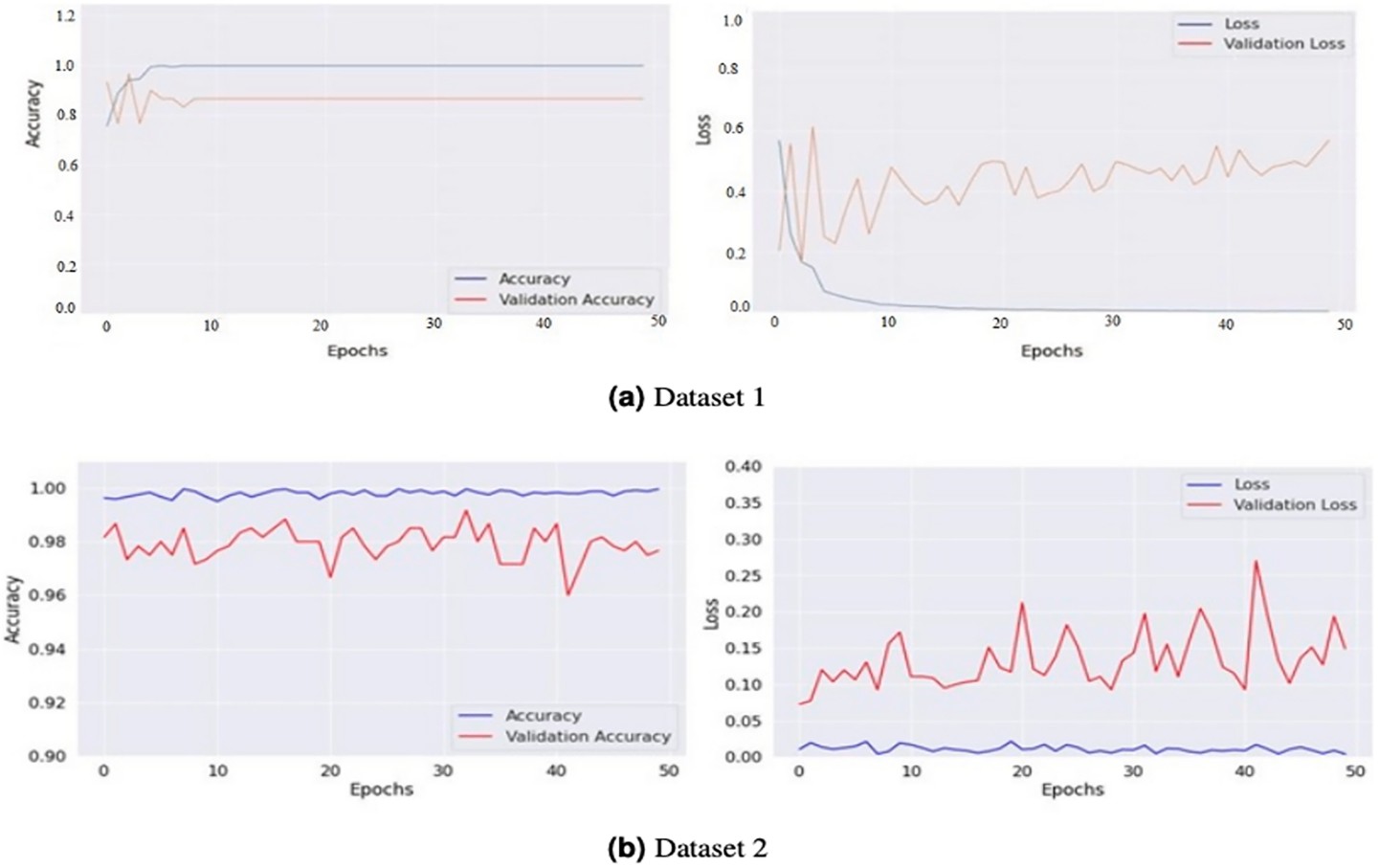

**Figure 5** Graphical representations of accuracy and loss for the training and validation processes for both Dataset 1 (A) and Dataset 2 (B). These graphs provide a visualization of the performance of the models during the training phase. The accuracy graphs show how well the model can classify the images correctly. In contrast, the loss graphs show how well the model can minimize the difference between the predicted and actual classes.

In Fig. 7, the density graph compares the distribution of real and generated images for both Dataset 1 and Dataset 2. The x-axis represents the pixel intensity values, while the y-axis represents the density of the images. The blue line represents the density of the real images, while the orange line represents the density of the generated images. In both datasets, the density of the generated images is very close to that of the real images, indicating that the generator can produce visually similar images in terms of their pixel intensity distribution. This indicates the effectiveness of the proposed CGAN approach in generating realistic brain tumor images.

## Qualitative and error analysis

To provide a more comprehensive understanding of our CGAN model's performance on Dataset 1 and Dataset 2, we have conducted a qualitative and error analysis in Tables 3 and

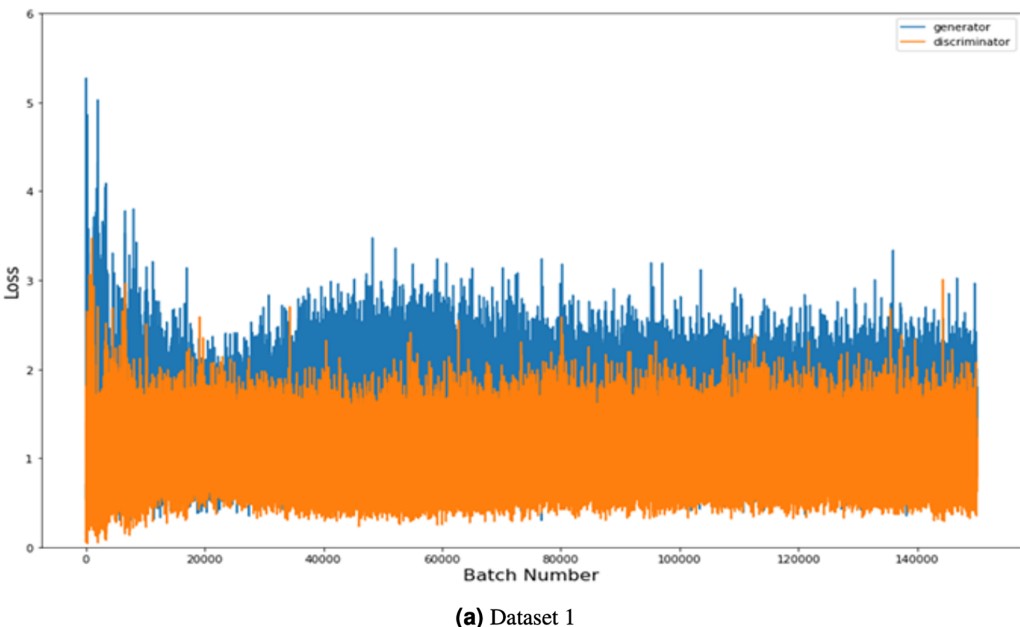

**(a)** Dataset 1

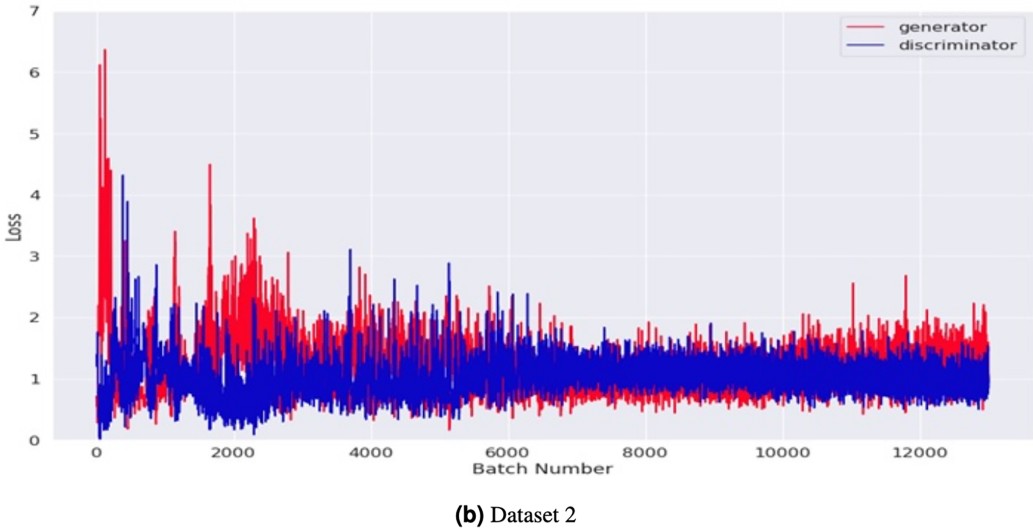

**(b)** Dataset 2

**Figure 6** **Performance comparison of generator and discriminator on the tumor training dataset.** The loss was measured concerning batch number for 8 batch sizes and 500 epochs with 100 noise dimensions.

4. This analysis aims to shed light on the model's behavior beyond quantitative metrics. Below, we present key findings from our qualitative and error analysis:

Dataset 1: Accuracy 93%, Class 'yes' (Brain Tumor Present): Precision: 0.90, Sensitivity: 1.00, Specificity: 1.00, and F1-score: 0.95. Class 'no' (No Brain Tumor): Precision: 1.00, Sensitivity: 0.83, Specificity: 1.00, and F1-score: 0.91.

In our qualitative analysis for Dataset 1, we examined instances where our CGAN model correctly classified brain tumor presence ('yes') and where it made errors ('no'). Visual examples from the dataset were reviewed to gain insights into these cases. We observed that while the model excelled in identifying brain tumor cases with high precision

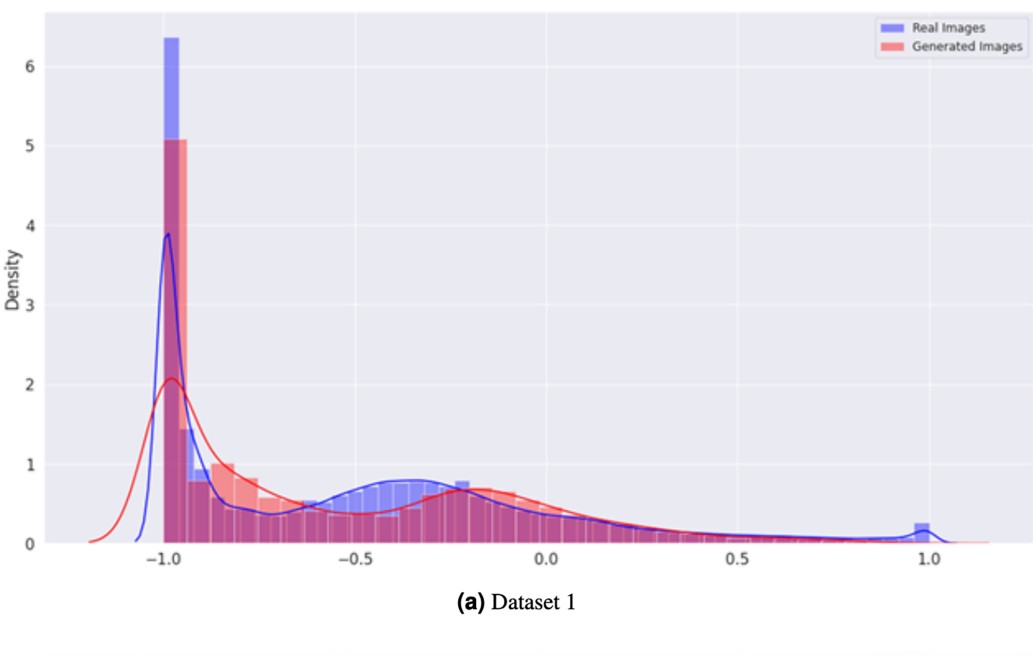

**(a)** Dataset 1

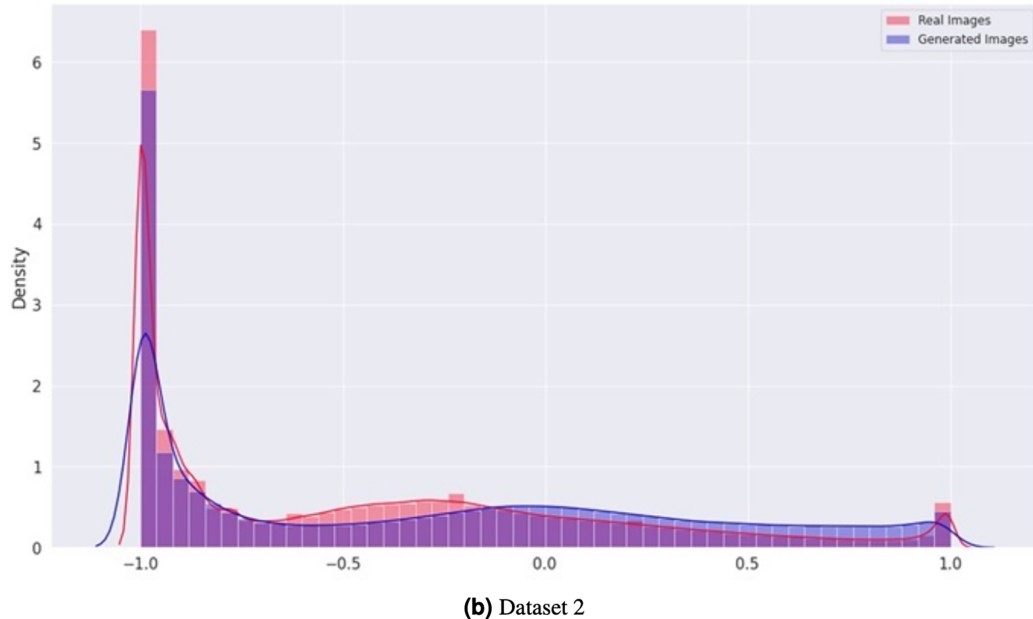

**(b)** Dataset 2

**Figure 7  Comparison of the density distribution of real and generated images for Datasets 1 and 2.**

**Table 3  CGAN model results on Dataset 1.**

| Class name | Precision | Sensitivity | Specificity | F1-score | Accuracy |
|---|---|---|---|---|---|
| Yes | 0.90 | 1.00 | 1.00 | 0.95 | 0.93 |
| No | 1.00 | 0.83 | 1.00 | 0.91 | |

**Table 4 CGAN model results on Dataset 2.**

| Class name | Precision | Sensitivity | Specificity | F1-score | Accuracy |
|---|---|---|---|---|---|
| Yes | 0.95 | 0.99 | 0.96 | 0.97 | 0.97 |
| No | 0.93 | 0.97 | 0.94 | 0.97 | |

and sensitivity, it encountered challenges in correctly classifying cases without tumors. Further investigation into these errors revealed that some of the false negatives could be attributed to subtle or ambiguous MRI images.

Dataset 2: Accuracy 97%, Class 'yes' (Brain Tumor Present): Precision: 0.95, Sensitivity: 0.99, Specificity: 0.96, F1-score: 0.97. Class 'no' (No Brain Tumor): Precision: 0.93, Sensitivity: 0.97, Specificity: 0.94, and F1-score: 0.97.

In the qualitative analysis for Dataset 2, we conducted a similar examination of model performance. Our model demonstrated strong precision, sensitivity, and accuracy in identifying brain tumor cases. However, we also noticed instances where the model incorrectly classified images without tumors. This prompted us to investigate the specific characteristics of these false positives.

Our qualitative and error analysis provides valuable insights into the model's strengths and limitations. We will include additional details, visual examples, and potential strategies for addressing these errors in the revised manuscript, enriching the discussion of our CGAN model's performance.

Table 5, provides a comparison of accuracy results for various GAN models used in the context of brain tumor classification. These models are evaluated for their effectiveness in accurately classifying brain tumors from medical images. Additionally, the table includes results for the proposed Conditional CGAN model applied to two different datasets, referred to as Dataset 1 and Dataset 2. The table consists of two columns: "Methodology" and "Accuracy." The "Methodology" column lists the different GAN-based models used for brain tumor classification, along with their respective references, providing readers with the sources for further study. The "Accuracy" column presents the accuracy scores achieved by each methodology, quantifying their performance in brain tumor segmentation.

Optimal DeepMRSeg GAN model proposed by *Neelima et al. (2022)* achieved an accuracy of 0.91. Deep GAN model proposed by *Ghassemi, Shoeibi & Rouhani (2020)* achieved an accuracy of 0.93. *Ahmad et al. (2022)* proposed VAEs-GAN model that was achieved an accuracy of 0.96. (*Asiri et al., 2023*) introduced MLD-GAN model that also achieved an accuracy of 0.96. Additionally, the Proposed CGAN model is presented in two rows, indicating its performance on two different datasets, labeled as "For Dataset 1" and "For Dataset 2." The accuracy achieved by the proposed CGAN model is recorded as 0.93 for Dataset 1 and an improved accuracy of 0.97 for Dataset 2.

This table serves as a reference for comparing the accuracy results of different GAN-based models in the context of brain tumor classification, with particular attention to the performance of the proposed CGAN model on two distinct datasets. It provides valuable

**Table 5 Comparison of accuracy results for existing GAN models and proposed CGAN model for brain tumor segmentation.**

| Methodology | Accuracy (%) |
| --- | --- |
| *Neelima et al. (2022)* | 0.91 |
| *Ghassemi, Shoeibi & Rouhani (2020)* | 0.93 |
| *Ahmad et al. (2022)* | 0.96 |
| *Asiri et al. (2023)* | 0.96 |
| Proposed CGAN (for Dataset 1) | 0.93 |
| Proposed CGAN (for Dataset 2) | 0.97 |

insights into the effectiveness of various methodologies in this critical medical imaging application.

## CONCLUSION

This study introduced a novel approach employing a conditional generative adversarial network (CGAN) fine-tuned with a convolutional neural network (CNN) to enhance the precision of brain tumor detection. The CGAN model was subjected to rigorous evaluation using publicly available brain tumor MRI image datasets (Dataset 1 and Dataset 2), each containing "yes" (indicating the presence of a brain tumor) and "no" (indicating no tumor) classes. The datasets underwent preprocessing, including resizing, noise removal, and standardization, ensuring a robust and consistent foundation for experimentation.The evaluation criteria encompassed essential statistical metrics, including precision, specificity, sensitivity, F1-score, and accuracy. Notably, our proposed CGAN model demonstrated exceptional performance, achieving an accuracy of 0.93 for Dataset 1 and an impressive accuracy of 0.97 for Dataset 2, surpassing existing techniques. Comparatively, the proposed CGAN model's accuracy outperformed several existing GAN-based models in the domain of brain tumor classification. Notably, it surpassed the accuracy achieved by the Optimal DeepMRSeg GAN, Deep GAN, VAEs-GAN, and MLD-GAN models, consolidating its effectiveness in this critical medical imaging application. our study introduces a novel and promising approach to enhance brain tumor detection using CGANs. The robust performance exhibited by our model on two distinct datasets highlights its potential for improving the accuracy and reliability of brain tumor diagnosis. The insights gained from this research contribute to the ongoing efforts to combat this life-threatening condition, offering hope for more accurate and timely diagnoses and ultimately saving lives.

### Funding

This work was supported by the Deanship of Scientific Research, Najran University, Kingdom of Saudi Arabia, under the research group funding program, grant code number

(NU/RG/MRC/12/10). The funders had no role in study design, data collection and analysis, decision to publish, or preparation of the manuscript.

## Grant Disclosures
The following grant information was disclosed by the authors:
Najran University: NU/RG/MRC/12/10.

## Competing Interests
The authors declare that they have no competing interests.

## Author Contributions
- Abdullah A. Asiri conceived and designed the experiments, performed the experiments, performed the computation work, prepared figures and/or tables, authored or reviewed drafts of the article, and approved the final draft.
- Muhammad Aamir performed the experiments, analyzed the data, performed the computation work, prepared figures and/or tables, and approved the final draft.
- Tariq Ali conceived and designed the experiments, performed the experiments, prepared figures and/or tables, authored or reviewed drafts of the article, and approved the final draft.
- Ahmad Shaf conceived and designed the experiments, performed the experiments, analyzed the data, performed the computation work, prepared figures and/or tables, and approved the final draft.
- Muhammad Irfan performed the experiments, prepared figures and/or tables, authored or reviewed drafts of the article, and approved the final draft.
- Khlood M. Mehdar analyzed the data, prepared figures and/or tables, authored or reviewed drafts of the article, and approved the final draft.
- Samar M. Alqhtani performed the experiments, performed the computation work, prepared figures and/or tables, authored or reviewed drafts of the article, and approved the final draft.
- Ali H. Alghamdi analyzed the data, performed the computation work, authored or reviewed drafts of the article, and approved the final draft.
- Abdullah Fahad A. Alshamrani analyzed the data, authored or reviewed drafts of the article, and approved the final draft.
- Osama M. Alshehri analyzed the data, authored or reviewed drafts of the article, and approved the final draft.

## Data Availability
The code is available at GitHub and Zenodo:
- https://github.com/iamshaf/CGAN.
- iamshaf. (2023). iamshaf/CGAN: CGAN-2023 (CGAN-2023). Zenodo. https://doi.org/10.5281/zenodo.8383887.

The dataset is publicly available at Kaggle:

- https://www.kaggle.com/datasets/navoneel/brain-mri-images-for-brain-tumor-detection.

- https://www.kaggle.com/datasets/ahmedhamada0/brain-tumor-detection.

## Supplemental Information

Supplemental information for this article can be found online at http://dx.doi.org/10.7717/peerj-cs.1667#supplemental-information.

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
