# Peer review of "Next-Gen brain tumor classification: pioneering with deep learning and fine-tuned conditional generative adversarial networks"

_PeerJ Computer Science, doi:10.7717/peerj-cs.1667_

## Round 0.1 · original submission · Major Revisions

Dear Authors,
Reviewers find merit in your manuscript, however, they recommended for significant revision. You are advised to address each and every comments of the reviewers, and revise your manuscript, including polishing of the English language.
Revision will be subjected to the re-review.
Good luck.

**Language Note:** The Academic Editor has identified that the English language must be improved. PeerJ can provide language editing services - please contact us at [email protected] for pricing (be sure to provide your manuscript number and title). Alternatively, you should make your own arrangements to improve the language quality and provide details in your response letter. – PeerJ Staff

Reviewer 1 ·

Basic reporting

1. The manuscript is about the Conditional Generative Adversarial Network (CGAN) that leverages the fine-tuning of CNN to achieve precise detection of brain tumors.

2. It is well-written and self-explanatory.

3. Related work papers should be organized in year-wise fashion (from previous years to 2023). Organized papers in different sub-sections (ML/DL-based approaches). Also, include current status and limitations of the state-of-the-art.

3. Figures-1 and 5(a) are not clear.

4. Equation - 6 (try to write in simple form of harmonic equation).

5. Table-5 (references missing).

Experimental design

1. Ablation study is missing.

2. Comparison with neural network baseline models is missing like CNN, LSTM, etc.

3. Comparison results in terms of F-score are missing in table-5.

4. Please share the GitHub code link in the paper.

5. Add “Qualitative and Error Analysis” of CGAN model as a new sub-section in experimental Section.

Validity of the findings

The main goal of this manuscript is to classify brain tumor. They proposed a model called CGAN. Evaluation results, tables, and figures need improvement.

Additional comments

: Major Revision is required.

Reviewer 2 ·

Basic reporting

The manuscript lacks proper English and it is highly advised to complete the proof read with professional services / fluent English support speakers.
The Literature Review is not well organised and State of Art Section should be upgraded to Literature Review in an organised way.
The figures submitted by the authors are not clear and as per the Journal Guidelines. The figures should be modified for clarity of vision and understanding.
Several terms in the Formal results do not include the definitions of the terms used in the equations and theorems. It is highly recommended to update these formulae with proper terms and meaning.

Experimental design

The research question focus on building a next generation brain tumor neural network. However it is found that the CGAN proposed by the author is not clearly explained in terms of the fresh conceptualisation. The proposed study seems to use the image processing using the python language, however, it lacks the novelty in research. The focus of the author should be more on proposing the new architecture and its salient features instead.
Investigations are done on two datasets from Kaggle, however it is recommended to use more better datasets / clinical data from authentic locations as :
https://figshare.com/articles/dataset/brain_tumor_dataset/1512427
The methods presented in the study need some serious changes in accordance with the proposed concept of CGAN. As an example D(Ri) and F(Ni) is not clearly mentioned. Their role and other specifications are not clear in the pseudocode. Several questions remained unanswered such as : Why the Sequential Model is used in the Working of the CGAN study? It should be included in the section of Model Working.

Validity of the findings

The datasets used in this study are from kaggle. However the authors are motivated to include results from the other datasets as:
https://www.ncbi.nlm.nih.gov/pmc/articles/PMC8061452/

The dataset used are clear but obsolete and legacy data. It is highly advised to the authors so that they can train the proposed study using some latest research data. The clinical analysis can be done and a simulation for the data should be made using the model for more accurate results.

Generation of fake images should be more clearly mentioned and the results for generation of these images must be made clear. The FP factor seems to be more studied and classification of the dataset can be done with tools like WEKA to make several classifiers work across the same. The results obtained then will be more beneficial.

Additional comments

Several new citations and reference should be included in the research study as:

https://levelup.gitconnected.com/brain-tumor-classification-using-tensorflow-and-transfer-learning-b4b7ab5fd7cf

https://www.mdpi.com/2079-9292/12/4/955


All the figures under results section must explain the findings and then comparison with the previous models used for similar kind of study as per the literature review.
The paper contains potential but with the aforesaid changes it can be more beneficial and acceptable.
Authors are advised to complete the reviews and modify the findings as per the above sections.

---

## Round 0.2 · accepted · Accept

The reviewers confirmed that all the suggested changes have been made. Reviewer 2 suggested improving the Quality of Fig 1. The manuscript may be accepted for publication in its current form.

Reviewer 1 ·

Basic reporting

Authors have addressed all my concerns. Manuscript seems acceptable in the present form.

Experimental design

Addressed all comments.

Validity of the findings

Addressed all comments.

Additional comments

Accept

Reviewer 2 ·

Basic reporting

Figure # 1 Still lacks the visibility.
Strongly recommended to upgrade it to vector graphics and submit again.

Experimental design

Much appreciated the updates.
However use of Clinical Datasets will be more beneficial.

Validity of the findings

No Comments!

Additional comments

Kindly modify the figure vector graphics quaility (especially for Fig 1)